# Trajectory inference across multiple conditions with condiments

Hector Roux de Bézieux [1,2], Koen Van den Berge [3,4,6], Kelly Street [5,7] & Sandrine Dudoit [1,2,3,7]

In single-cell RNA sequencing (scRNA-Seq), gene expression is assessed individually for each cell, allowing the investigation of developmental processes, such as embryogenesis and cellular differentiation and regeneration, at unprecedented resolution. In such dynamic biological systems, cellular states form a continuum, e.g., for the differentiation of stem cells into mature cell types. This process is often represented via a trajectory in a reduced-dimensional representation of the scRNA-Seq dataset. While many methods have been suggested for trajectory inference, it is often unclear how to handle multiple biological groups or conditions, e.g., inferring and comparing the differentiation trajectories of wild-type and knock-out stem cell populations. In this manuscript, we present **condiments**, a method for the inference and downstream interpretation of cell trajectories across multiple conditions. Our framework allows the interpretation of differences between conditions at the trajectory, cell population, and gene expression levels. We start by integrating datasets from multiple conditions into a single trajectory. By comparing the cell's conditions along the trajectory's path, we can detect large-scale changes, indicative of differential progression or fate selection. We also demonstrate how to detect subtler changes by finding genes that exhibit different behaviors between these conditions along a differentiation path.

The emergence of RNA sequencing at the single-cell level (scRNA-Seq) has enabled a new degree of resolution in the study of cellular processes. The ability to consider biological processes as a continuous transition of cell states instead of individual discrete stages has permitted a finer and more comprehensive understanding of dynamic processes such as embryogenesis and cellular differentiation. Trajectory inference (TI) was one of the first applications that leveraged this continuum[1] and a consequential number of methods have been proposed since then[2–4]. Saelens et al.[5] offer an extensive overview and comparison of such methods. Analysis of scRNA-Seq datasets using a curated database reveals that about half

of all datasets were used for trajectory inference[6]. At its core, TI represents a dynamic process as a directed graph. Distinct paths along this graph are called lineages. Individual cells are then projected onto these lineages and – given a root state (either user-provided or automatically detected) – their distance along each path is called pseudotime. In this setting, developmental processes are often represented in a tree structure, while cell cycles are represented as a loop. Following TI, other methods have been proposed to investigate differential expression (DE) along or between lineages, either as parts of TI methods[3,7] or as separate modules that can be combined to create a full pipeline[8].

[1]Division of Biostatistics, School of Public Health, University of California, Berkeley, CA, USA. [2]Center for Computational Biology, University of California, Berkeley, CA, USA. [3]Department of Statistics, University of California, Berkeley, CA, USA. [4]Department of Applied Mathematics, Computer Science and Statistics, Ghent University, Ghent, Belgium. [5]Division of Biostatistics, Department of Population and Public Health Sciences, Keck School of Medicine of USC, Los Angeles, CA, USA. [6]Present address: Statistics and Decision Sciences, J&J Innovative Medicine, Beerse, Belgium. [7]These authors contributed equally: Kelly Street, Sandrine Dudoit. ✉e-mail: kelly.street@usc.edu; sandrine@stat.berkeley.edu

Spearheaded by technological and laboratory developments allowing increased throughput of scRNA-Seq studies, recent experiments study dynamic systems affected by different experimental perturbations or biological conditions. This includes, for example, situations where a biological process is studied both under a normal (or control) condition and under an intervention such as a treatment[9–11] or a genetic modification[12]. In other instances, one may want to contrast healthy versus diseased[13] cells or even more than two conditions[14]. In steady-state systems, a reasonable approach consists of clustering the cells into biologically relevant cell identities and assessing differential abundance, i.e., imbalances between conditions, for each cluster. However, in dynamic systems, clustering the cells and assessing condition imbalances for each cluster ignores the continuity of the cellular state transition process, making these approaches suboptimal. For example, borrowing from the field of mass cytometry[15], **milo**[16] and **DAseq**[17] rely on a common low-dimensional representation of all observations and define data-driven local neighborhoods in which they test for differences in compositions. Each of these methods shows clear improvements in performance over cluster-based methods, which assess imbalances between conditions for each cell state/type cluster, and provide a more principled approach that better reflects the nature of the system.

However, many studies with multiple conditions actually involve processes that can be described by a trajectory. Utilizing this underlying biology could increase both the interpretability of the results and the ability to detect true and meaningful changes between conditions. In this manuscript, we present the **condiments** workflow, a general framework to analyze dynamic processes under multiple conditions that leverages the concept of a trajectory structure. While **condiments** has a more specific focus than **milo** or **DAseq**, it compensates for this by improving the quality of the differential abundance assessment with better-performing tests and simplifying and enhancing the biological interpretation by breaking down the comparison into several smaller and pertinent questions. Our proposed analysis workflow is divided into three steps. In Step 1, **condiments** considers the trajectory inference question, assessing whether the dynamic process is fundamentally different between conditions, which we call *differential topology*. In Step 2, it tests for global differences between conditions, both along lineages – *differential progression* – and between lineages – *differential fate selection*. Lastly, in Step 3, it estimates gene expression profiles similarly to Van den Berge et al.[8] and tests whether gene expression patterns differ between conditions along lineages, therefore extending the scope of *differential expression* and improving on cluster-based methods such as MAST[18], scde[18], or padoga2[19].

In this manuscript, we first present the **condiments** workflow, by detailing the underlying statistical model and providing an explanation and intuition for each step. We then benchmark **condiments** against more general methods that test for differential abundance to showcase how leveraging the existence of a trajectory improves the assessment of differential abundance. Finally, we demonstrate the flexibility and improved interpretability of the **condiments** workflow in three case studies that span a variety of biological settings and topologies.

## Results

### General model and workflow

To help visualize the workflow, we first simulated toy datasets that illustrate different scenarios that we will later encounter in our case studies. In Fig. 1, we have four examples, one per column, with two conditions each: a control condition and a knock-out (KO) condition. The first example represents a setting where there are no differences at all between conditions. As we can see in the reduced-dimensional representation (first row), the orange and blue cells are similarly distributed. Therefore, we observe no differences, whether differential topology, differential progression, or differential fate selection. In the second and third examples, cells differentiate along the same structure so we have no differential topology. In the second example, however, we can see that there are nearly no orange cells at the beginning of the trajectory and that cells in the KO condition progress faster: this is an example of differential progression. In the third example, there are nearly no orange cells after the bifurcation in the bottom lineage and cells in the KO condition select the top lineage preferentially compared to the control: this is an example of differential fate selection. Note that,

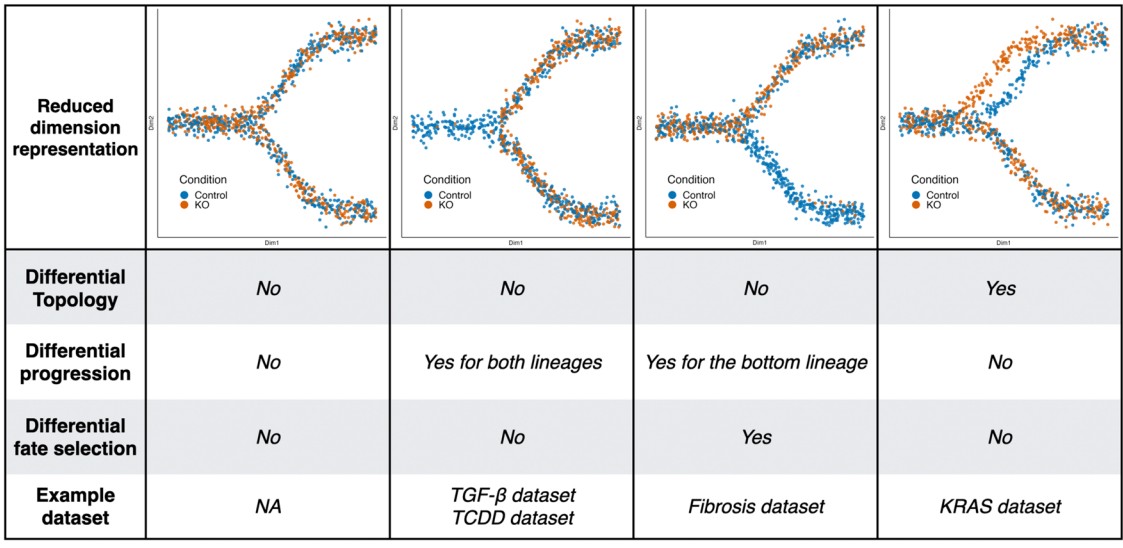

| | Column 1 | Column 2 | Column 3 | Column 4 |
|---|---|---|---|---|
| **Reduced dimension representation** | *(scatter plot, Control/KO)* | *(scatter plot, Control/KO)* | *(scatter plot, Control/KO)* | *(scatter plot, Control/KO)* |
| **Differential Topology** | No | No | No | Yes |
| **Differential progression** | No | Yes for both lineages | Yes for the bottom lineage | No |
| **Differential fate selection** | No | No | Yes | No |
| **Example dataset** | NA | TGF-β dataset, TCDD dataset | Fibrosis dataset | KRAS dataset |

**Fig. 1 | Illustrating the first two steps of the condiments workflow with several scenarios.** We have four different toy datasets, one per column, with their reduced-dimensional representation displayed in the first row. Each dataset contains 1, 000 cells equally distributed between two conditions: control in blue and knock-out (KO) in orange. They represent possible outcomes from the first two steps of the **condiments** workflow. The first dataset has no differences between the two conditions. In the second dataset, the KO accelerates the differentiation process: we

therefore have differential progression, but neither differential topology nor differential fate selection. In the third dataset, the KO blocks differentiation in the bottom lineage: we therefore have differential fate selection and differential progression, but no differential topology. Finally, in the fourth dataset, the KO modifies the top lineage: we therefore have differential topology. The last row of the table indicates case studies corresponding to each of the scenarios.

since lineages are not distinguishable before they bifurcate, we only see differential progression in the bottom lineage since orange cells only progress along that lineage until the bifurcation. In the fourth example, we can see that the top lineage is altered in the KO condition: this is an example of differential topology and we would infer separate trajectories. We can still compare the lineages and, after correcting for the differential topology, we see no differential progression or differential fate selection. Case studies described in a later section will give real-world examples of similar settings.

We now describe the data structure and statistical model. All mathematical symbols used in this section are recapitulated in Table 1. We observe gene expression measures for $J$ genes in $n$ cells, resulting in a $J \times n$ count matrix $\mathbf{Y}$. For each cell $i$, we also know its condition label $c(i) \in \{1, ..., C\}$ (e.g.,"treatment" or "control", "knock-out" or "wild-type"). We assume that, for each condition $c$, there is an underlying developmental structure $\mathcal{T}_c$, or trajectory, that possesses a set of $L_c$ lineages.

For a given cell $i$ with condition $c(i)$, its position along the developmental path $\mathcal{T}_{c(i)}$ is defined by two vectors: a vector of $L_{c(i)}$ pseudotimes $\mathbf{T}_i$, which represent how far the cell has progressed along each lineage; and a unit-norm vector of $L_{c(i)}$ weights $\mathbf{W}_i$ ($\|\mathbf{W}_i\|_1 = 1$), which represents how likely it is that the cell belongs to each lineage. That is, for each cell $i$, there is one pseudotime and one weight per lineage, with:

$$\mathbf{T}_i \sim G_{c(i)} \text{ and } \mathbf{W}_i \sim H_{c(i)}. \tag{1}$$

The (multivariate) cumulative distribution functions (CDF) $G_c$ and $H_c$ are condition-specific and we make limited assumptions on their properties (see the Methods section for details). Using this notation, we can properly define a trajectory inference (TI) method as a function that takes as input $\mathbf{Y}$ – and potentially other arguments (e.g., cell cluster labels) – and returns estimates of $L_c$, $\mathbf{T}$, and $\mathbf{W}$ through the estimate for $\mathcal{T}_c$. That is, methods such as slingshot[2] or monocle3[20] use the count matrix to estimate the number of lineages, as well as the pseudotime and lineage assignments (or weights) of each cell.

The first question to ask in our workflow (**Step 1**) is: Should we fit a common trajectory to all cells regardless of their condition? Or are the developmental trajectories too dissimilar between conditions? To demonstrate what this means, consider two extremes. For a dataset that consists of a mix of bone marrow stem cells and epithelial stem cells, using tissue as our condition, it is obvious that the developmental trajectories of the two conditions are not identical and should be estimated separately. On the other hand, if we consider a dataset where only a few genes are differentially expressed between conditions, the impact on the developmental process will be minimal and it is sensible to estimate a single common trajectory.

We favor fitting a common trajectory for several reasons. Firstly, fitting a common trajectory is a more stable procedure since more cells are used to infer the trajectory. Secondly, our workflow still provides ways to test for differences between conditions even if a common trajectory is inferred. In particular, fitting a common trajectory between conditions does not require that cells of distinct conditions differentiate similarly along that trajectory. Finally, fitting different trajectories greatly complicates downstream analyses since we may need to map between distinct developmental structures in order to be able to compare them (e.g., each lineage in the first trajectory must match exactly one lineage in the second trajectory). Therefore, our workflow recommends fitting a common trajectory if the differences between conditions are small enough.

To quantify what small enough is, we rely on two approaches. The first is a visual diagnostic tool called imbalance score. It requires as input a reduced-dimensional representation $\mathbf{X}$ of the data $\mathbf{Y}$ and the condition labels. Each cell is assigned a score that measures the imbalance between the local and global distributions of condition labels. Similarly to Dann et al.[16,21], the neighborhood of a cell is defined using a $k$-nearest neighbor graph on $\mathbf{X}$, which allows the method to scale very well to large values of $n$. Cell-level scores are then locally

## Table 1 | Notation

| Symbol | Description |
|---|---|
| $C$ | The number of conditions in a dataset. |
| $c(i)$ | The condition label of the $i^{th}$ cell. |
| $G_c$ | Cumulative distribution function for the pseudotimes of a trajectory under condition $c$. It represents how cells are distributed along the different lineages. |
| $H_c$ | Cumulative distribution function for the weights of a trajectory under condition $c$. It represents how cells are distributed between the different lineages. |
| $J$ | The number of genes in a scRNA-Seq dataset. |
| $L$ | The number of lineages in a trajectory. |
| $n$ | The number of cells in a scRNA-Seq dataset. |
| $s_{jlc}$ | The smoother that represents the gene expression pattern along lineage $l$ for gene $j$ in condition $c$. It is a smooth function estimated by tradeSeq. |
| $\mathbf{T}_i$ | The pseudotime for a cell $i$. For a trajectory with more than one lineage, this is a vector, with one value per lineage. It measures how far the cell has progressed along each lineage. |
| $\mathcal{T}_c$ | The structure of a trajectory under condition $c$. It is what trajectory inference methods such as slingshot or monocle3 are trying to identify. |
| $\mathbf{W}_i$ | The weights for a cell $i$. For a trajectory with more than one lineage, this is a vector, with one value per lineage. It measures how close each cell is to each lineage. A weight of 1 means the cell belongs only to that lineage; a weight of 0 means the cell does not belong to that lineage. |
| $\mathbf{X}$ | A reduced-dimensional representation of the dataset. For example, for UMAP, this is a matrix with two columns, where each row gives the 2D coordinates of a cell in that reduced dimension. |
| $\mathbf{Y}$ | The count matrix. Each cell represents the expression level. of a gene (row-wise) in a cell (column-wise). |

The table provides the symbol, as well as a short explanation of what it represents. Symbols are listed in alpha-numerical order.

scaled using smoothers in the reduced-dimensional space (see the Methods section).

However, a visual representation of the scores may not always be sufficient to decide whether or not to fit a common trajectory in less obvious cases. Therefore, we introduce a more quantitative approach, the `topologyTest`. This test assesses whether we can reject the following null hypothesis:

$$H_0 : \forall (c_1, c_2) \in \{1, \ldots, C\}^2, \mathcal{T}_{c_1} = \mathcal{T}_{c_2}. \tag{2}$$

We can test hypothesis (2) using the `topologyTest`, which involves comparing the actual distribution of pseudotimes for condition-specific trajectories to their distribution when condition labels are permuted (see the Methods section for details). Since we want to favor fitting a common trajectory and we only want to discover differences that are not only statistically significant but also biologically relevant, the tests typically include a minimum magnitude requirement for considering the difference between distributions to be significant (similar to a minimum log-fold-change for assessing DE). Examples of results for the `topologyTest` can be seen in Fig. 1. In the first column, there is no differential topology between the blue and orange condition, while in the fourth column, the top lineage is different between the two conditions.

In practice, the `topologyTest` requires maintaining a mapping between each of the trajectories, both between conditions and between permutations (see the Methods section where we define a mapping precisely). Trajectory inference remains a semi-supervised task, that generally cannot be fully automated. In particular, the number of estimated lineages might change between different permutations for a given condition, precluding a mapping. As such, the `topologyTest` is only compatible with certain TI methods that allow for the specification of an underlying skeleton structure, including slingshot and TSCAN[2,4]. More details and practical implementation considerations are discussed in the Methods section.

For Steps 2 and 3, we are no longer limited to specific TI methods. Any TI method can be used as input. We can then ask whether cells from different conditions behave similarly as they progress along the trajectory.

The workflow then focuses on differences between conditions at the trajectory level (**Step 2**) and asks: What are the global differences between conditions? To facilitate the interpretation of the results, we break this into two separate sub-questions.

Although the topology might be common, cells from different conditions might progress at different rates along the trajectory (**Step 2a**). For example, a treatment might limit the cells' differentiation potential compared to the control, or instead speed it up. In the first case, one would expect to have more cells at the early stages and fewer at the terminal state, when comparing treatment and control. In the second column of Fig. 1, we can see another example: there are more orange cells at the latest stages of the trajectory, compared to the blue cells. Using our statistical framework, testing for differential progression amounts to testing equality of the pseudotime distributions between conditions:

$$H_0 : \forall (c_1, c_2) \in \{1, \ldots, C\}^2, G_{c_1} = G_{c_2}. \tag{3}$$

This test can also be conducted at the individual-lineage level, by comparing univariate distributions.

In order to assess the null hypotheses in the `progressionTest`, we rely on non-parametric tests to compare two or more distributions, e.g., the Kolmogorov-Smirnov test[22] or the classifier test[23]. More details and practical implementation considerations are discussed in the Methods section.

Although the topology might be common, cells in different conditions might also differentiate in varying proportions between the

lineages (**Step 2b**). For example, an intervention might lead to cells selecting one lineage over another, compared to the control condition, or might alter survival rates of differentiated cells between two end states. In the third column of Fig. 1, we can see another example: there are nearly no orange cells at the later stage of the bottom lineage of the trajectory, compared to the blue cells. Cells in the orange condition are more likely to end up in the top lineage. In all examples above, the weight distribution will be different between the control and treatment. Assessing differential fate selection at the global level amounts to testing, in our statistical framework, the null hypothesis of equal weight distributions between conditions

$$H_0 : \forall (c_1, c_2) \in \{1, \ldots, C\}^2, H_{c_1} = H_{c_2}. \tag{4}$$

The above null hypotheses can again be tested by relying on non-parametric test statistics. We also discuss specific details and practical implementation in the Methods section. This test can also be conducted for a single pair of lineages $(l, l')$.

The `progressionTest` and `fateSelectionTest` are quite linked and will therefore often return similar results, as Fig. 1 shows. However, they do answer somewhat different questions. In particular, looking at single-lineage (`progressionTest`) and lineage-pair (`fateSelectionTest`) test statistics will allow for a better understanding of the global differences between conditions. Differential fate selection does not necessarily imply differential progression and vice versa. The simulations will show some examples of this.

Steps 1 and 2 focus on differences at a global level (i.e., aggregated over all genes) and will detect large changes between conditions. However, such major changes are usually ultimately driven by underlying differences in gene expression patterns (**Step 3**). Furthermore, even in the absence of global differences, conditions might still have a more subtle impact at the gene level. In the third step, we therefore compare gene expression patterns between conditions for each of the lineages.

Following the `tradeSeq` manuscript by Van den Berge et al.[8], we consider a general and flexible model for gene expression, where the gene expression measure $Y_{ji}$ for gene $j$ in cell $i$ is modeled with the negative binomial generalized additive model (NB-GAM) described in Equation (11). We extend the `tradeSeq` model by additionally estimating condition-specific average gene expression profiles for each gene. We therefore rely on lineage-specific, gene-specific, and condition-specific smoothers, $s_{jlc}$.

With this notation, we can introduce the `conditionTest`, which, for a given gene $j$, tests the null hypothesis that these smoothers are identical across conditions:

$$H_0 : s_{jlc_1} = s_{jlc_2}, \forall (c_1, c_2), \forall l. \tag{5}$$

As in tradeSeq, we rely on the Wald test to test $H_0$ in terms of the smoothers' regression coefficients. We can also use the fitted smoothers to visualize condition-specific gene expression or cluster genes according to their expression patterns.

## Simulations
We generate multiple trajectories using the dyngen simulation framework provided by Cannoodt et al.[24]. Within this framework, it is possible to knock out a specific gene. Here, we knock out a master regulator that drives differentiation into the second lineage. The strength of this knock-out can be controlled via a multiplier parameter $m$: If $m = 1$, there is no effect; if $m = 0$, the knock-out is total; if $0 < m < 1$, we have partial knock-out; if $m > 1$, the master regulator is over-expressed and cells differentiate much faster along the second lineage.

Four types of datasets are generated: a) simple branching trajectories (two lineages, e.g., Fig. 2a) of 3, 500 cells, with equal parts wild-type and knock-out; b) trajectories with two consecutive branchings

(and thus three lineages, e.g., Fig. 2b) of 3, 500 cells, with equal parts wild-type and knock-out; c) branching trajectories (two lineages) of 5, 000 cells with three conditions, wild-type, knock-out with multiplier $m$, and induction with multiplier $1/m$ (Fig. 2c); d) more complex trajectories (five lineages, e.g, Fig. 2d) of 3, 500 cells, with equal parts wild-type and knock-out.

Since the simulation framework cannot generate trajectories with distinct topologies for the different conditions, we start the **condiments** workflow at Step 2. We use either slingshot or monocle3 upstream of **condiments**. We compare the `progressionTest` and `fateSelectionTest` to methods that also do not rely on clustering, but instead take into account the continuum of differentiation. **milo**[16] and **DAseq**[17] both define local neighborhoods using $k$-nearest neighbors graphs. They then look at differences of condition proportions in these neighborhoods to test for what they call differential abundance. These methods return multiple tests per dataset (i.e., one per neighborhood), so we adjust for multiple hypothesis testing using the Benjamini-Hochberg procedure for control of the false discovery rate (FDR)[25]. By applying **milo**, **DAseq**, and **condiments** on the simulated datasets, we can compare the results of the tests versus the values of $m$:

We count a true positive when a test rejects the null and $m \neq 1$, and a true negative when the test fails to reject the null and $m = 1$. Note that since monocle3 performs hard-assignments of cells to lineages, the weight distributions are degenerate, and thus we cannot run the `fateSelectionTest` downstream of this method.

We compare the methods' ability to detect correct differences between conditions using five measures: The true negative rate (TNR), positive predictive value (PPV), true positive rate (TPR), negative predictive value (NPV), and F1-score, when controlling the FDR at a nominal level of 5%. More details on the simulation scenarios and performance measures can be found in the Methods section. Results are displayed in Fig. 2e.

On all simulations, all methods display strong results for the TNR and PPV. However, the performances for the TPR (power), NPV, and F1-rate vary quite widely. On the datasets with two or three lineages, the `progressionTest`, downstream of either slingshot or monocle3, performs the best, followed by the `fateSelectionTest`, DAseq,—**DAseq**, and **milo**. On the third simulation setting with three conditions, we cannot benchmark **DAseq** since its testing framework is restricted to two conditions. **DAseq** slightly outperforms the

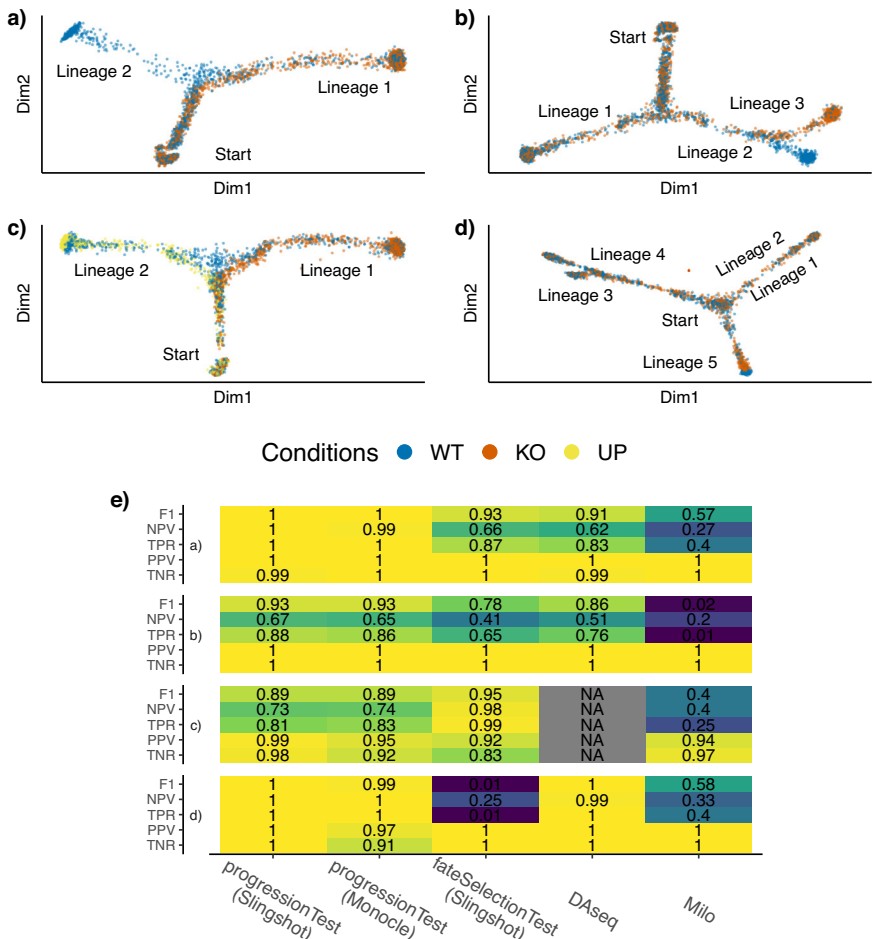

**Fig. 2 | Simulated datasets.** Four types of datasets are generated, with respectively 2, 3, 2, and 5 lineages, and 2, 2, 3, and 2 conditions (WT - wild type, KO - knock-out, UP - upregulated). Reduced-dimensional representations of these datasets are presented in panels (**a–d**). Fig. S6 shows the same plot as (**d**), using dimensions 3 and 4, showing that Lineages 1 and 2 do separate. After generating multiple versions of the datasets for a range of values of $m$, representing more or less differences between conditions, we compare the performance of the `progressionTest` after running either slingshot or monocle3; the `fateSelectionTest` after running slingshot; **DAseq**[17] and **milo**[16], when controlling the false discovery rate at nominal levels of 5% using the Benjamini-Hochberg[25] procedure. In (**e**), each cell of the table represents a performance measure associated with one test on one of the four types of dataset, where the blue-to-yellow color palette corresponds to low-to-high performance. Cells are also colored according to the performance measure. Overall, the `progressionTest` and `fateSelectionTest` work well across all datasets. **DAseq** also has good performances with two conditions, but cannot be extended to more. Exact simulation parameters and performance measures are specified in the Methods section. The poor performance of the `fateSelectionTest` for (**d**) is expected, since the effect only appears at the end of Lineage 5, not at the branching itself.

`progressionTest` on the dataset with 5 lineages although the results are quite similar. On those datasets, the knock-out only affects the tip of lineage 5 but not the proportion of cells that select that lineage. Therefore, the fact that the `fateSelectionTest` has no power (.01) is expected. This displays the increased flexibility and interpretability procured by the **condiments** workflow, while still outperforming competitors.

The dyngen framework does not generate trajectories with differential topology, therefore the correct workflow would be to fit a common trajectory. We can nonetheless study the impact of fitting separate trajectories on the rest of the workflow. We therefore consider two settings: correctly fitting a common trajectory or incorrectly fitting separate trajectories, to represent the two possible outcomes of Step 1. We then compare the performance of the `progressionTest` and `fateSelectionTest` in the two settings. The test performances are identical for the `progressionTest`, and only slightly impacted in the `fateSelectionTest` case, when looking at a receiver operating characteristic (ROC) plot (see Fig. S5). The lower performance when incorrectly fitting separate trajectories may stem from the fact we tend to have more false positives, which can be expected: fitting separate trajectories when it is not needed amounts to over-fitting to the data.

Even if the correct decision is taken after Step 1, the trajectory inference may still be imperfect. We therefore assess the performance of the `progressionTest` as we add increasing noise to the true simulated pseudotimes and lineage assignment values. Since the dyngen framework assumes hard lineage assignments, we cannot test the `fateSelectionTest`. With up to 60% of cells improperly assigned (noise of .3) and 200% noise on pseudotime (see the Methods section for details), the `progressionTest` still correctly rejects the null at the 5% nominal significance level, showing that moderate errors in the trajectory inference have small impact on the test outcome. See Fig. S4.

This is also reflected in the previous simulations of Fig. 2. While slingshot correctly identifies the proper number of lineages 98% of the time, monocle3 only does so 54% of the time. This is probably because the former allows for the specification of both start and endpoints, while the latter only allows for a start point. Nonetheless, the performance of the `progressionTest` following monocle3 is nearly identical to that downstream of slingshot.

We consider four real datasets as case studies for the application of the **condiments** workflow. Table 2 gives an overview of these datasets and summary results. These case studies aim to demonstrate the versatility and usefulness of the **condiments** workflow, as well as showcase how to interpret and use the tests in practice. The results for the TGF-β dataset and the Fibrosis dataset are presented below while the results for the TCDD and KRAS datasets are presented in Supplementary Results. Preprocessing for all three datasets is described in Supplementary Methods[26–28].

### TGF-β dataset

McFaline-Figueroa et al.[9] studied the epithelial-to-mesenchymal transition (EMT), where cells migrate from the epithelium (inner part of the tissue culture dish) to the mesenchyme (outer part of the tissue culture dish) during development. The developmental process therefore is both temporal and spatial. Moreover, the authors studied this system under two settings: a mock (control) condition and a condition under activation of transforming growth factor β (TGF-β).

After pre-processing, normalization, and integration (see Supplementary Methods S1.3), we have a dataset of 9,268 cells, of which 5,207 are mock and 4,241 are TGF-β-activated. The dataset is represented in reduced dimension using UMAP[29] (Fig. 3a). Adding the spatial label of the cells (Fig. 3b) shows that the reduced-dimensional representation of the gene expression data captures the differentiation process.

We can then run the **condiments** workflow, beginning with the differential topology question. The imbalance score of each cell is computed and displayed in Fig. 3c. Although some regions do display strong imbalance in conditions, there is no specific pattern along the developmental path. This is confirmed when we run the `topologyTest`, which has a nominal $p$-value of 0.38. We clearly fail to reject the null hypothesis and we consequently fit a common trajectory to both conditions using slingshot with the two spatial labels as clusters. The resulting single-lineage trajectory is shown in Fig. 3d.

Next, we ask whether the TGF-β treatment impacts the differentiation speed. The developmental stage of each cell is estimated using its pseudotime. Plotting the per-condition kernel density estimates of the pseudotime distributions in Fig. 3e reveals a strong treatment effect. The pseudotime distribution for the mock cells is trimodal, likely reflecting initial, intermediary, and terminal states. By contrast, the initial mode is not present in the TGF-β condition, and the second is skewed towards higher pseudotime values. This is very consistent with the fact that the treatment is a growth factor that would stimulate differentiation, as shown in the original publication and confirmed in the literature on EMT[30]. Testing for equality of the two distributions with the `progressionTest` confirms the visual interpretation. The nominal $p$-value associated with the test is smaller than $2.2 \times 10^{-16}$ and we reject the null that the distributions are identical. Since the trajectory is limited to one lineage, the `fateSelectionTest` is not applicable.

Then, we proceed to identifying genes whose expression patterns differ between the mock and TGF-β conditions. After gene filtering, we fit smoothers to 10,549 genes, relying on the model described in Equation (11). We test whether the smoothers are significantly different between conditions using the `conditionTest`. Testing against a log-fold-change threshold of 2, we find 1,993 genes that are dynamically differentially expressed between the two conditions when controlling the false discovery rate at a nominal level of 5%. Figure 3f shows the two genes with the highest Wald test statistic. The first gene, *LAMC2*, was also found to be differentially expressed in the original publication and has been shown to regulate EMT[31]. The second gene, *TGFBI* or TGF-β-induced gene, is not surprising, and was also labelled as differentially expressed in the original publication. The DE genes confirm known biology around TGF-β signaling and EMT. For example, it is known that TGF-β signaling occurs through a complex of TGF-β receptor 1

**Table 2 | Summary of all case studies datasets**

| Dataset | $n$ | $C$ | $L$ | topologyTest | progressionTest | fateSelectionTest | DE |
|---|---|---|---|---|---|---|---|
| TGF-β[9] | 9268 | 2 | 1 | 0.38 | $\leq 2.2 \times 10^{-16}$ | *NA* | 1,993 |
| Fibrosis[34] | 14,462 | 2 | 2 | 1 | $\leq 2.2 \times 10^{-16}$ | $\leq 2.2 \times 10^{-16}$ | 3 |
| TCDD[10] | 9951 | 2 | 1 | 0.07 | $\leq 2.2 \times 10^{-16}$ | *NA* | 2,144 |
| KRAS[49] | 10,177 | 3 | 3 | $\leq 2.2 \times 10^{-16}$ | $\leq 2.2 \times 10^{-16}$ | $\leq 2.2 \times 10^{-16}$ | 363 |

The table reports the name, number of cells $n$, number of conditions $C$, number of lineages $L$ for each dataset, as well as the $p$-value resulting from testing for differential topology, progression, and fate selection, and the number of differentially expressed genes between conditions according to the `conditionTest`. Note that the `fateSelectionTest` cannot be run on a dataset with only one lineage. No adjustment was made for multiple testing.

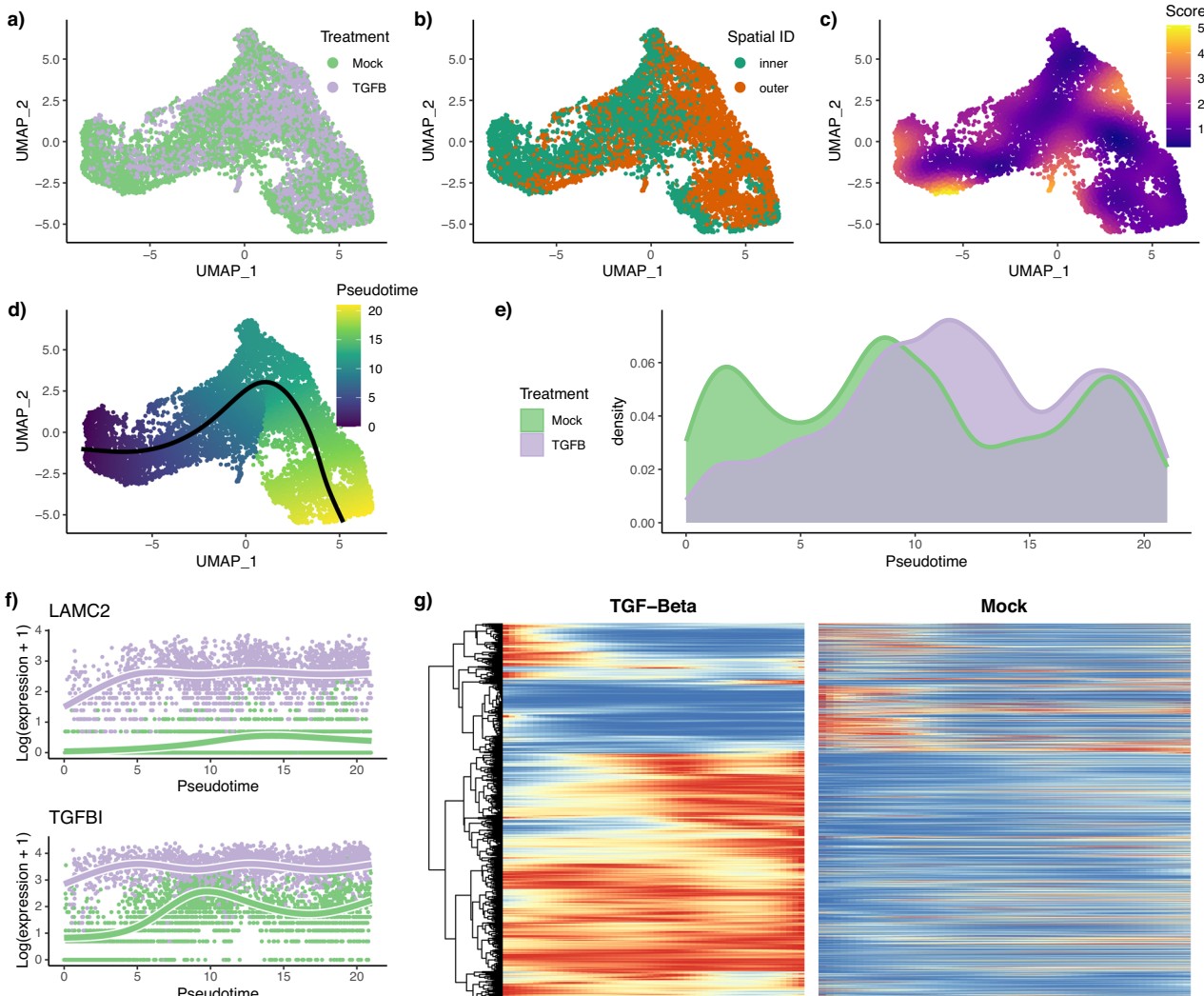

**Fig. 3 | TGF-$\beta$ dataset.** Full workflow. After normalization and projection on a reduced-dimensional space (using UMAP), the cells can be colored either by treatment label (**a**) or spatial origin (**b**). Using the treatment label and the reduced-dimensional coordinates, an imbalance score is computed and displayed (**c**). The `topologyTest` fails to reject the null hypothesis of no differential topology, and a common trajectory is therefore fitted and cells colored according to pseudotime (**d**). However, there is differential progression between conditions, as indicated by different pseudotime distributions along the trajectory (**e**), and we reject the null using the `progressionTest`. The tradeSeq gene expression model is fitted using the trajectory inferred by slingshot. Differential expression between conditions is assessed using the `conditionTest` and genes are ranked according to their Wald test statistic. The expression measures and fitted values as a function of pseudo-time are displayed for the genes with the two highest test statistics (**f**). After adjusting $p$-values of the `conditionTest` to control the FDR at a nominal level of 5%, we display expression fitted values for the DE genes for both conditions using a pseudocolor image, where fitted values are scaled to a $[0, 1]$ range for each gene (blue-to-red color palette represents low-to-high expression) (**g**).

(TGFBR1) and TGF-$\beta$ receptor 2 (TGFBR2)[32], and both genes are found to be significantly upregulated after TGF-$\beta$ treatment (Fig. S10). Looking at all 1,906 DE genes, we can cluster and display their expression patterns along the lineage for both conditions (Fig. 3g and identify several groups of genes that have different patterns between the two conditions. These groups of genes may be further explored to better understand TGF-$\beta$ induced EMT. For example, genes that are differentially expressed later in the EMT may be further downstream targets of TGF-$\beta$ signaling.

Finally, we perform a gene set enrichment analysis on all the genes that are differentially expressed between the conditions using fgsea[33]. We find two significant gene sets that differentiate treatment and control: biological adhesion and locomotion. This is in full con-cordance with the biology: the TGF-$\beta$ treatment accelerates the migration and differentiation of cells on the tissue culture dish. Indeed, the TGF-$\beta$-accelerated EMT requires epithelial cells to disband inter-cellular junctions and subsequently migrate to the mesenchyme.

We also reproduce the analysis using monocle3 as a trajectory inference method. We cannot perform Step 1 with monocle3. We use the results from Step1 with slingshot and fit a common trajectory: monocle3 also finds a single lineage developmental path (Fig. 4a). We can then test for differential progression and find the same result as with slingshot: We have clear differential progression ($p$-value of $2.2 \times 10^{-16}$). Likewise, we find that the Wald statistics from the `con-ditionTest` are very highly correlated (Pearson correlation coeffi-cient of 0.97) when performed downstream of either slingshot or monocle3. Finally, using the same FDR nominal level as before, we find 1613 DE genes downstream of monocle3, 92% of which were also deemed DE downstream of slingshot. Therefore, changing the trajec-tory inference method has no strong impact on the downstream analysis.

We can also compare the results produced by **DAseq** and **milo** on this dataset. Both methods find similar regions of differential abun-dance (Fig. 4b, c), but the results are hard to interpret; they are not

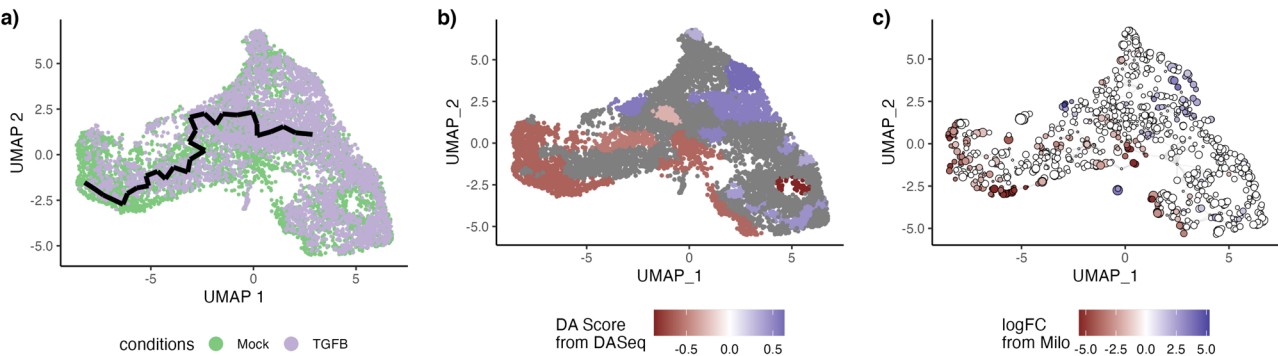

**Fig. 4 | TGF-β dataset.** Alternate analyses. Using monocle3 to infer the trajectory also yields a single-lineage trajectory (**a**), albeit different than the one inferred by slingshot. After inferring the trajectory with monocle3, there is also differential progression: the pseudotime distributions along the trajectory are not identical and we indeed reject the null hypothesis using the `progressionTest`. On the other hand, while **DAseq** (**b**) and **milo** (**c**) correctly identify that there is differential abundance, their results are much harder to interpret.

linked to the trajectory, so biological interpretation of the statistical result is more challenging.

### Fibrosis dataset

Habermann et al.[34] investigated chronic interstitial lung diseases (ILD) by sequencing 10 non-fibrotic control lungs and 20 ILD lungs. In its later stage, the disease progresses to pulmonary fibrosis, associated with epithelial tissue remodelling. The original paper uses scRNA-Seq to investigate how this remodelling plays among many cell types. Here, we focus on a subset of the data comprised of 14, 462 cells, with 5, 405 control and 9, 057 ILD cells. These data underlie Fig. 3 of the original paper and focus on two cell lineages, starting from, respectively, AT2 and SCGB3A2+ cells, that each differentiate independently into AT1 cells.

We use the pre-processed and normalized data from the original paper to obtain the reduced-dimensional representation depicted in Fig. 5a (colored by disease status) and Fig. 5b (colored by cell type as defined in the original work). The reduced-dimensional representation captures the differentiation process, with both AT2 cells and SCGB3A2+ cells differentiating into AT1 cells through an intermediate 'transitory AT2' stage. Moreover, the SCGB3A2+ cell type contains nearly no control cells.

To dive more deeply into the differences between control and ILD cells, we run the **condiments** workflow. The imbalance score, displayed in Fig. 5c, strongly validates the initial observation: The AT2-AT1 lineage has relatively low imbalance scores while the SCGB3A2+ cluster shows strong imbalance. Interestingly, the transitional AT2 cluster shows no particular local imbalance, suggesting that it is indeed a common stage for both lineages, as the original paper demonstrated.

We then run the `topologyTest` downstream of slingshot (see the Methods section). On the full dataset, the test's p-value is $3.0 \times 10^{-12}$: estimating the lineages from each condition separately produces trajectories that are far more divergent than what can be produced from permutation. This is expected, as only 3% of the cells in the SCGB3A2+ cluster come from the control group. When the trajectory is estimated with only the control cells, the SCGB3A2+ lineage is spuriously drawn towards the AT2 cells (see Fig. S11).

However, when running the `topologyTest` after removing the SCGB3A2+ cells (and therefore only focusing on the second lineage), we fail to reject the null (p-value of 1). This common lineage can therefore be estimated using both conditions, while the first is only present in the treatment condition. We can therefore fit a common trajectory to the full dataset, and we find the two lineages depicted in Fig. 5d.

We can then move to Step 2 of the **condiments** workflow. The `fateSelectionTest` gives clear results: its p-value is $\leq 2.2 \times 10^{-16}$. When we look at the distribution of lineage weights in Fig. 5e, we can see three clear peaks: at 0 (the cell does not belong to that lineage), .5 (the cell belongs to both lineages), and 1 (the cell only belongs to that lineage). Because the sum of weights over the two lineages equals 1 for each cell, the distribution of the weights along Lineage 2 is symmetrical to the distribution along Lineage 1. We can clearly see that, for the control condition, we have some cells that belong to both lineages (weights of .5), but most cells belong only to Lineage 2 (weights of 0 for Lineage 1 and of 1 for Lineage 2). The `fateSelectionTest` provides a principled way of assessing this. Note that the test is named based on a setting where cells differentiate from a single root stage into multiple endpoints. Here, since we have the reverse (multiple root stages progression to a similar endpoint), it would be more appropriate to speak of differential origins. Looking at distributions of pseudotimes reinforces the fact that few cells in control belong to lineage 1. On the first lineage, most of the control cells are at the beginning of the lineage while the ILD cells are more evenly distributed, with a strong concentration at the end (see Fig. 5f). On the second (shared) lineage, there is some differential progression, but much less. Concordingly, the `progressionTest`'s p-value is $\leq 2.2 \times 10^{-16}$.

We can then proceed on to Step 3 and look for genes with different expression patterns between conditions, within a lineage. We fit the smoothers to 10, 100 genes. Testing against a log-fold-change threshold of 2, we find only 3 genes that are differentially expressed between conditions, according to the `conditionTest`, when controlling the FDR at a nominal level of 5%. This result might seem surprising at first but it is expected. Indeed, if we had a reverse scenario with a single root point and a branching into two lineages, we would expect many genes to be differentially expressed between conditions, thus driving the global differences we observed in Step 2. Here, however, all global differences come from the fact that there are only a handful of control cells in the SCGB3A2+ stage. Once we account for this, cells differentiate similarly. In the disease state, the AT2 to AT1 differentiation path is unchanged compared to the normal state; the only difference is that AT1 cells can also originate from a new cell state (SCGB3A2+).

To compare the two possible origins of AT1 cells, we can look for genes that have strong dynamic differences between lineages, regardless of condition. Using the `patternTest` from tradeSeq, we find 1, 589 DE genes when controlling the FDR at a nominal level of 5%. The gene with the strongest difference between the two lineages is the *SCGB3A2* gene (Fig. S12a). This gene was also listed in the original paper, as well as 3 other genes: *SFTPC*, *ABCA3*, and *AGER* (see Fig. S12b–d). While our workflow confirms all four candidate genes as DE genes, we additionally identify many more to be further investigated. To further explore the relevance of these DE genes, we perform a gene set enrichment analysis on all the genes that are differentially expressed between the lineages. The top enriched gene sets include

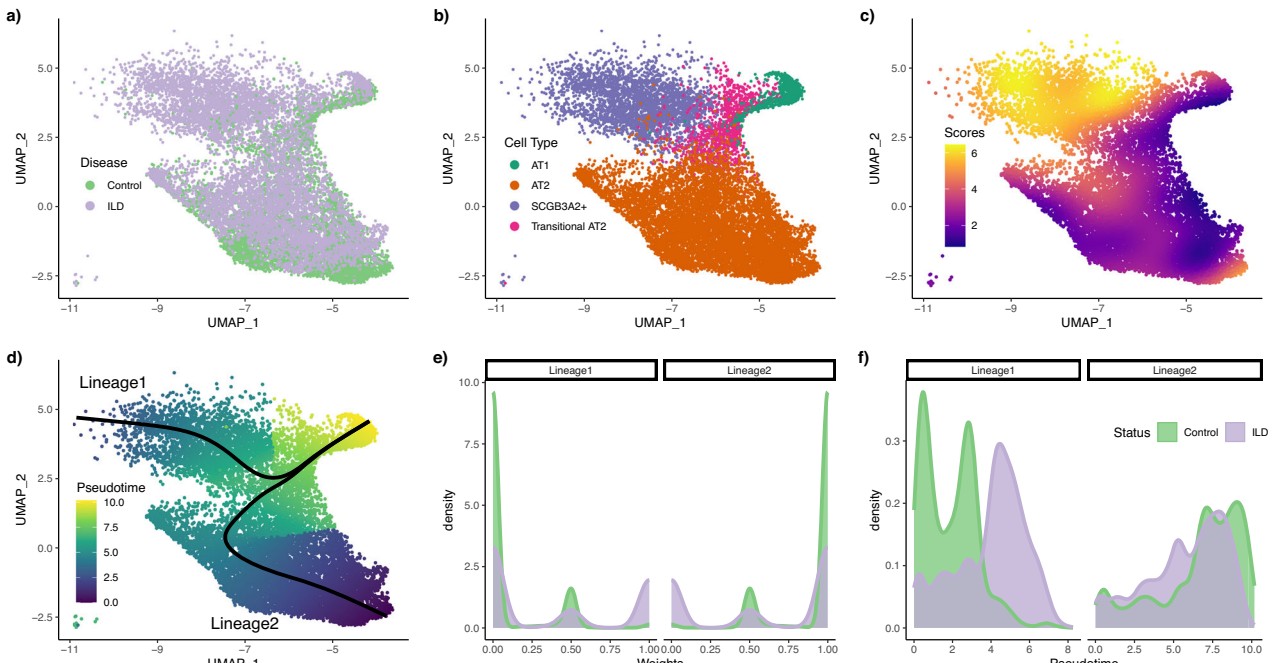

**Fig. 5 | Fibrosis dataset.** Step 1 and 2. After normalization and projection on a reduced-dimensional space (using UMAP), the cells can be colored either by disease status (**a**) or cell type (**b**). Using the disease status and the reduced-dimensional coordinates, an imbalance score is computed and displayed (**c**). The `topologyTest` fails to reject the null hypothesis of no differential topology and a common trajectory is therefore fitted (**d**). However, there is both differential fate selection and differential progression between conditions. The weight distributions (**e**) and the pseudotime distributions (**f**) are not identical between conditions, and we indeed reject the nulls using, respectively, the `fateSelectionTest` and the `progressionTest`.

adhesion, defense response, cell proliferation, epithelium development and morphogenesis, as well as immune response (Supplementary Table S1); biological processes that are all relevant to the system being studied.

## Discussion

In this manuscript, we have introduced **condiments**, a full workflow to analyze dynamic systems under multiple conditions. By separating the analysis into several steps, **condiments** offers a flexible framework with increased interpretability. Indeed, we follow a natural progression through a top-down approach, by first studying overall between-condition differences in trajectories with the `topologyTest`, then differences in abundance at the trajectory level with the `progressionTest` and `fateSelectionTest`, and finally gene-level differences in expression with the `conditionTest`.

As demonstrated in the simulation studies, taking into account the dynamic nature of biological systems via the trajectory representation enables **condiments** to better detect true changes between conditions. The flexibility offered by our implementation, which provides multiple non-parametric tests for comparing distributions, also allows us to investigate a wide array of scenarios. This is evident in the four case studies presented in the manuscript. Indeed, in the TGF-β case study we have a developmental system under treatment and control conditions, while in the Fibrosis case study we compare a normal system and its disease counterpart. In the TCDD case study, the continuum does not represent a developmental process but rather spatial separation. In the KRAS case study, the conditions do not reflect different treatments but instead different cancer models. This shows that **condiments** can be used to analyze a wide range of datasets.

Taking batch effects into account can be difficult, particularly as the different conditions often also represent different batches. Indeed, some interventions cannot be delivered on a cell-by-cell basis and this creates unavoidable confounding between batches and conditions. Normalization and integration of the datasets must therefore be done without eliminating the underlying biological signal. This balance can be hard to strike, as discussed in Zhao et al.[17]. Proper experimental design – such as having several batches per condition – or limiting batch effects as much as possible – for example, sequencing a mix of conditions together – can help lessen this impact. In settings where there is no confounding of batches and conditions, **condiments** offers additional tools to better perform batch correction. Still, some amount of confounding is sometimes inherent to the nature of the problem under study.

Normalization can still remove most of the batch effects before performing trajectory inference. If the normalization is insufficient, this might lead to rejecting the null in the `topologyTest`, as seen in the KRAS case study. In that setting, it is still possible to perform the downstream analysis: fitting one trajectory per condition is in fact a way to account for the remaining batch effects and therefore solve the issue for Step 2. Finally, when fitting smoothers to gene expression measures in Step 3, it is possible to add covariates such as batch, if batches are not fully confounded with conditions.

The tests used in the workflow (e.g., Kolmogorov-Smirnov test) assume that the pseudotime and weight vectors are known and independent random variables for each cell. However, this is not the case; they are estimated using TI methods which use all samples to infer the trajectory, and each estimate inherently has some uncertainty. Here, we ignore this dependence, as is the case in other differential abundance methods, which assume that the reduced-dimensional coordinates are observed independent random variables even though they are being estimated using the full dataset. We stress that, rather than attaching strong probabilistic interpretations to *p*-values (which, as in most RNA-Seq applications, would involve a variety of hard-to-verify assumptions and would not necessarily add much value to the analysis), we view the *p*-values produced by the **condiments** workflow as useful numerical summaries for guiding the decision to fit a common trajectory or condition-specific trajectories and for exploring trajectories across conditions and identifying genes for further inspection.

Other methods have been presented that can generate $p$-values with probabilistic interpretation[35,36]. However, current methods are restricted to only one lineage, or scale poorly beyond a thousand cells, showing the need for further work in this domain.

Splitting the data into two groups, where the first is used to estimate the trajectory and the second is used for pseudotime and weight estimation could, in theory, alleviate the dependence issue, at the cost of smaller sample sizes. However, this would ignore the fact that, in practice, users perform exploratory steps using the full data before performing the final integration, dimensionality reduction, and trajectory inference. Moreover, results on simulations show that all methods considered keep excellent control of the false discovery rate despite the violation of the independence assumptions. This issue of "double-dipping" therefore seems to have a limited impact in practice.

The two issues raised in the previous paragraphs highlight the need for independent benchmarking. Simulation frameworks such as dyngen[24] are crucial. They also need to be complemented by real-world case studies, which will become easier as more and more datasets that study dynamic systems under multiple conditions are being published. **condiments** has thus been developed to be a general and flexible workflow that will be of use to researchers asking complex and ever-changing questions.

## Methods

### Tests for equality of distributions
Consider a set of $n$ i.i.d. observations, $\mathbf{X}$, with $\mathbf{X}_i \sim \mathbf{P}_1$, and a second set of $m$ i.i.d. observations, $\mathbf{Y}$, with $\mathbf{Y}_j \sim \mathbf{P}_2$, independent from $\mathbf{X}$. For example, in our setting, $\mathbf{X}$ and $\mathbf{Y}$ may represent estimated pseudotimes for cells from two different conditions. We limit ourselves to the case where $\mathbf{X}$ and $\mathbf{Y}$ are random vectors of the same dimension $d$.

The general goal is to test the null hypothesis that $\mathbf{X}$ and $\mathbf{Y}$ have the same distribution, i.e., $H_0 : \mathbf{P}_1 = \mathbf{P}_2$.

### The two-sample and weighted Kolmogorov-Smirnov test
Consider the case where $\mathbf{X}$ and $\mathbf{Y}$ are scalar random variables (i.e., $d = 1$). The associated empirical cumulative distribution functions (ECDF) are denoted, respectively, by $\mathbf{F}_{1,n}$ and $\mathbf{F}_{2,m}$. The univariate case occurs, for example, when there is only one lineage in the trajectory(ies), so that the pseudotime estimates are scalars.

In this setting, one can test $H_0$ using the standard Kolmogorov-Smirnov test[22], with test statistic defined as:

$$\mathbf{D}_{n,m} \equiv \sup_x |\mathbf{F}_{1,n}(x) - \mathbf{F}_{2,m}(x)|.$$

The rejection region at nominal level $\alpha$ is

$$\left[ \sqrt{-\frac{1}{2} \times \log \frac{\alpha}{2} \times \frac{n+m}{n \times m}}, \infty \right).$$

That is, we reject the null hypothesis at the $\alpha$-level if and only if $\mathbf{D}_{n,m} \geq \sqrt{-1/2 \times \log \alpha/2 \times \frac{n+m}{n \times m}}$.

We can also consider a more general setting where we have weights $w_{1,i} \in [0, 1]$ and $w_{2,j} \in [0, 1]$ for each of the observations, $i = 1, \ldots, n$ and $j = 1, \ldots, m$. In trajectory inference, the weights may denote the probability that a cell belongs to a particular lineage in the trajectory. Following Monahan[37], we modify the Kolmogorov-Smirnov test in two ways. Firstly, the empirical cumulative distribution functions are modified to account for the weights

$$\mathbf{F}_{1,n}(x) = \frac{1}{\sum_{i=1}^{n} w_{1,i}} \sum_{i=1}^{n} w_{1,i} \times \mathcal{I}_{(-\infty, x]}(\mathbf{X}_i)$$

$$\mathbf{F}_{2,m}(x) = \frac{1}{\sum_{j=1}^{m} w_{2,j}} \sum_{j=1}^{m} w_{2,j} \times \mathcal{I}_{(-\infty, x]}(\mathbf{Y}_j).$$

Secondly, the definition of $\mathbf{D}_{n,m}$ is unchanged, but the significance threshold is updated, that is, the rejection region is

$$\left[ \sqrt{-\frac{1}{2} \times \log \frac{\alpha}{2} \times \frac{n'+m'}{n' \times m'}}, \infty \right),$$

where

$$n' = \frac{\left( \sum_{i=1}^{n} w_{1,i} \right)^2}{\sum_{i=1}^{n} w_{1,i}^2} \text{ and } m' = \frac{\left( \sum_{j=1}^{m} w_{2,j} \right)^2}{\sum_{j=1}^{m} w_{2,j}^2}.$$

### The multivariate classifier test
Suppose that we have a binary classifier $\delta(\cdot)$, which could be, for example, a multinomial regression or SVM classifier. This classifier is a function from the support of $\mathbf{X}$ and $\mathbf{Y}$ into $\{1, 2\}$. The data are first split into a learning and a test set, such that the test set contains $n_{\text{test}}$ observations, equally-drawn from each population, i.e., there are $n_{\text{test}}/2$ observations $\mathbf{X}^{(\text{test})}$ from $\mathbf{X}$ and $n_{\text{test}}/2$ observations $\mathbf{Y}^{(\text{test})}$ from $\mathbf{Y}$. Next, the classifier is trained on the learning set. We denote by $Acc \equiv |\{i : \delta(\mathbf{X}_i^{(\text{test})}) = 1\} \bigcup \{j : \delta(\mathbf{Y}_j^{(\text{test})}) = 2\}|$ the number of correct assignations made by the classifier on the test set.

If $n = m$, under the null hypothesis of identical distributions, no classifier will be able to perform better on the test set than a random assignment would, i.e., where the predicted label is a *Bernoulli*(1/2) random variable. Therefore, testing the equality of the distributions of $\mathbf{X}$ and $\mathbf{Y}$ can be formulated as testing

$$H_0 : \mathbb{E}[Acc] = \frac{n_{\text{test}}}{2} \text{ vs. } H_1 : \mathbb{E}[Acc] > \frac{n_{\text{test}}}{2}.$$

Under the null hypothesis, the distribution of $Acc$ is:

$$Acc \sim_{H_0} Binom(n_{\text{test}}, 1/2).$$

As detailed in Lopez-Paz and Oquab[23], one can use the classifier to devise a test that will guarantee the control of the Type 1 error rate.

In practice, we make no assumptions about the way in which the distributions we want to compare might differ, which means the classifier needs to be quite flexible. Following Lopez-Paz and Oquab[23], we chose to use either a $k$-nearest-neighbor classifier ($k$-NN) or a Random Forests classifier[38], since such classifiers are fast and flexible. Hyper-parameters are chosen through cross-validation on the learning set. To avoid issues with class imbalance, we downsample the distribution with the largest number of samples first so that each distribution has the same number of observations. That is, we have $n' = \min(m, n)$ observations in each condition (or class). A fraction (by default 30%, user-defined) is kept as test data, so that $n_{\text{test}} = 0.3 \times n'$. We then train the classifier on the learning data, and select the tuning parameters through cross-validation on that learning set. Finally, we predict the labels on the test set and compute the accuracy of the classifier on that test set. This yields our classifier test statistic.

It is interesting to note that the classifier test is valid no matter the classifier chosen. However, the choice of classifier will have obvious impact on the power of the test.

### Other multivariate methods
Although we have found that the classifier test performs best in practice, there are many methods that test for the equality of two multivariate distributions. We have implemented a few such methods in **condiments**, in case users would like to try them: The two-sample kernel test[39], the permutation test relying on the Wasserstein distance[40,41], or the distinct method[42]. (see descriptions in the Supplementary Methods S1.1).

## Extending the setting to more than two distributions

Consider $C \geq 2$ sets of samples, such that, for $c \in \{1, \ldots, C\}$, we have $n_c$ i.i.d. observations $\mathbf{X}^{(c)}$ with $\mathbf{X}_i^{(c)} \sim \mathbf{P}_c$. We want to test the null hypothesis:

$$H_0 : \mathbf{P}_{c_1} = \mathbf{P}_{c_2}, \forall c_1, c_2 \in \{1, \ldots, C\} \text{ and } c_1 \neq c_2.$$

While extensions of the Kolmogorov-Smirnov test[43] and the two-sample kernel test[44] have been proposed, we choose to focus only on the framework that is most easily extended to $C$ conditions, namely, the classifier test. Indeed, the $C$-condition classifier test requires choosing a multiple-class classifier instead of a binary classifier (which is the case for the $k$-NN classifier and Random Forests), selecting $n_{\text{test}}/C$ observations for each class in the test set, and testing:

$$H_0 : \mathbf{E}[Acc] = \frac{n_{\text{test}}}{C} \text{ vs. } H_1 : \mathbf{E}[Acc] > \frac{n_{\text{test}}}{C}.$$

Under the null hypothesis, the distribution of $Acc$ is:

$$Acc \sim_{H_0} Binom(n_{\text{test}}, 1/C).$$

## Extending the setting by considering an effect size

The null hypothesis of the (weighted) Kolmogorov-Smirnov test is $H_0 : P_1 = P_2$. We can modify this null hypothesis by considering an effect size threshold $t$, such that $H_0(t) : \sup_x |P_1(x) - P_2(x)| \leq t$. The test statistic is then modified as:

$$\mathbf{D}'_{n,m} \equiv \max(\mathbf{D}_{n,m} - t, 0)$$

and the remainder of the testing procedure is left unchanged.

Similarly, the null and alternative hypotheses of the classifier test can be modified to test against an effect size threshold $t$ as follows

$$H_0 : \mathbb{E}[Acc] \leq \frac{n_{\text{test}}}{C} + t \text{ vs. } H_1 : \mathbb{E}[Acc] > \frac{n_{\text{test}}}{C} + t.$$

## General statistical model for the trajectories

Consider a set of condition labels $c \in \{1, \ldots, C\}$ (e.g.,"treatment" or "control", "knock-out" or "wild-type"). For each condition, there is a given topology/trajectory $\mathcal{T}_c$ that underlies the developmental process. This topology is generally in the form of a tree, with a starting state which then differentiates along one or more lineages; but one can also have a circular graph, e.g., for the cell cycle. In general, a trajectory is defined as a directed graph.

We denote by $L_c$ the number of unique paths – or lineages – in the trajectory $\mathcal{T}_c$. For example, for a tree structure, paths go from the root node (stem cell type) to the leaf nodes (differentiated cell type). For a cell cycle, any node can be be used as the start. A cell $i$ from condition $c(i)$ is characterized by the following random variables:

$$\mathbf{T}_i \sim G_{c(i)} : \text{A vector of pseudotimes, one per lineage of } \mathcal{T}_{c(i)}$$

$$\mathbf{W}_i \sim H_{c(i)} : \text{A vector of weights, one per lineage of } \mathcal{T}_{c(i)}, \text{ s.t. } ||\mathbf{W}_i||_1 = 1.$$

Note that the distribution functions are condition-specific. We further make the following assumptions:

- All $G_c$ and $H_c$ distributions are continuous;
- The support of $G_c$ is bounded in $\mathbb{R}^{L_c}$ for each $c$;
- The support of $H_c$ is $[0,1]^{L_c}$ for each $c$.

The gene expression model will be discussed below, in the Testing for differential expression section.

Many algorithms have been developed to infer lineages from single-cell data[5]. Most algorithms provide a binary indicator of lineage assignment, that is, the $\mathbf{W}_i$ vectors are composed of 0s and 1s, so that a cell either belongs to a lineage or it does not. (Note that when cells fall along a lineage prior to a branching event, this vector may include multiple 1s, violating our constraint that the $\mathbf{W}_i$ have unit norm. In such cases, we normalize the weights to sum to 1).

## Mapping between trajectories

Many of the tests that we introduce below assume that the cells from different conditions follow "similar" trajectories. In practice, this means that we either have a common trajectory for all conditions or that there is a possible mapping from one lineage to another. The term "mapping" is more rigorously defined as follows.

**Definition 1.** The trajectories $\{\mathcal{T}_c : c \in \{1, \ldots, C\}\}$ have a mapping if and only if $\forall (c_1, c_2) \in \{1, \ldots, C\}^2$, $\mathcal{T}_{c_1}$ and $\mathcal{T}_{c_2}$ are isomorphic.

If there is a mapping, this implies in particular that the number of lineages $L_c$ per trajectory $\mathcal{T}_c$ is the same across all conditions $c$ and we call this this common value $L$. Since a graph is always isomorphic with itself, a common trajectory is a special case of a situation where there is a mapping.

**Definition 2.** The trajectories $\{\mathcal{T}_c : c \in \{1, \ldots, C\}\}$ have a partial mapping if and only if $\forall (c_1, c_2) \in \{1, \ldots, C\}^2$, there exist subgraphs $\mathcal{T}'_{c_1} \subset \mathcal{T}_{c_1}$ and $\mathcal{T}'_{c_2} \subset \mathcal{T}_{c_2}$ that are isomorphic.

Essentially, this means that the changes induced by the various conditions do not disturb the topology of the original trajectory *too much*. The above mathematical definitions aim to formalize what *too much* is. Indeed, if the conditions lead to very drastic changes, it will be quite obvious that the trajectories are different and comparing them will mostly be either non-informative or will not require a complex framework. We aim to build a test that retains reasonable power in more subtle cases.

## Imbalance score

Consider a set of $n$ cells, with associated condition labels $c(i) \in \{1, \ldots, C\}$ and coordinate vectors $\mathbf{X}_i$ in $d$ dimensions, usually corresponding to a reduced-dimensional representation of the expression data obtained via PCA or UMAP[29,45], $i = 1, \ldots, n$.

Let $\mathbf{p} = \{p_c\}_{c \in \{1, \ldots, C\}}$ denote the "global" distribution of cell conditions, where $p_c$ is the overall proportion of cells with label $c$ in the sample of size $n$. The imbalance score of a cell reflects the deviation of the "local" distribution of conditions in a neighborhood of that cell compared to the global distribution $\mathbf{p}$. Specifically, for each cell $i$, we compute its $k$-nearest-neighbor graph using the Euclidean distance in the reduced-dimensional space. We therefore have a set of $k$ neighbors and a set of associated neighbor condition labels $c_{i,\kappa}$ for $\kappa \in \{1, \ldots, k\}$. We then assign to the cell a $z$-score, based on the multinomial test statistic $P(\{c_{i,\kappa}\}_{\kappa \in \{1, \ldots, k\}}, \mathbf{p})$, as defined in supplementary section S1.2. Finally, we smooth the $z$-scores in the reduced-dimensional space by fitting $s$ cubic splines for each dimension. The fitted values for each of the cells are the imbalance scores. Thus, the imbalance scores rely on two user-defined parameters, $k$ and $s$. We set default values of 10 for both parameters. However, since this is meant to be an exploratory tool, we encourage users to try different values for these tuning parameters and observe the effect on the imbalance scores to better understand their data.

## Differential topology

The imbalance score only provides an exploratory visual assessment of local imbalances in the distribution of cell conditions. However, we need a more global and formal way to test for differences in topology between condition-specific trajectories. That is, we wish to test the null

hypothesis

$$H_0 : \mathcal{T}_{c_1} = \mathcal{T}_{c_2}, \forall (c_1, c_2) \in \{1, \ldots, C\}^2. \tag{6}$$

In practice, in order to test $H_0$, we have a set of cells $i$ with condition labels $c(i)$. Not all trajectory inference methods define $\mathcal{T}$ in the same way, but they all do estimate pseudotimes. We can thus estimate the pseudotimes of each cell when fitting a trajectory for each condition. We then want to compare this distribution of pseudotimes to a null distribution of pseudotimes corresponding to a common topology. To generate this null distribution, we use permutations in the following manner:

a. Estimate $\mathbf{T}_i$ for each $i$ by inferring one trajectory per condition, using any trajectory inference method.
b. Randomly permute the condition labels $c(i)$ to obtain new labels $c(i)'$, re-estimate $\mathbf{T}'_i$ for each $i$.
c. Repeat the permutation $r$ times (by default, $r = 100$).

Under the null hypothesis of a common trajectory, the $n\mathbf{T}_i$ should therefore be drawn from the same distribution as the $r \times n\mathbf{T}'_i$. We can test this using the weighted Kolmogorov-Smirnov test (if $L = 1$), the kernel two-sample test (if $C = 2$), or the classifier test (any $C$). This is the `topologyTest`.

The aforementioned tests require that the samples be independent between the distributions under comparison. However, here, the two distributions correspond to different pseudotime estimates for the same cells so the samples are not independent. Even within distributions, the independence assumption is violated, as the pseudotimes are estimated using trajectory inference methods that rely on all samples. Moreover, within the $\mathbf{T}'_i$, we have $r$ pseudotime estimates of each cell.

The first two violations of the assumptions are hard to avoid and are further addressed in the Discussion section. However, we can eliminate the third one by simply taking the average $\mathbf{T}'_i$ for each cell. We then compare two distributions each with $n$ samples. Both options (with and without averaging) are implemented in the **condiments** R package, but the default is the average.

Under the null, there should exist a mapping between trajectories, both within conditions and between permutations. However, in practice, most trajectory inference methods will be too unstable to allow for automatic mapping between the runs. Indeed, they might find a different number of lineages for some runs. Moreover, even if the number of lineages and graph structure remained the same across all permutations, mapping between permutations would break even more the independence assumption since the condition labels would need to be used.

Therefore, for now, the `topologyTest` test is limited to two trajectory inference methods, slingshot[2] and TSCAN[4], where a set graph structure can be prespecified. Both methods rely on constructing a minimum spanning tree (MST) on the centroids of input clusters in a reduced-dimensional space to model branching lineages. In TSCAN, a cell's pseudotime along a lineage is determined by its projection onto a particular branch of the tree and its weight of assignment is determined by its distance from the branch. slingshot additionally fits simultaneous principal curves. A cell's pseudotime along a lineage is determined by its projection onto a particular curve and its weight of assignment is determined by its distance from the curve.

We therefore construct the MST on the full dataset (i.e., using all the cells regardless of their condition label), based on user-defined cluster labels. Then, we keep the same graph structure as input to either TI method: the nodes are the centers of the clusters, but restrained to cells of a given condition. This way, the path and graph structure are preserved. Note, however, that there is no guarantee that the graph remains the MST when it is used for TI on a subset of cells.

In the examples from Fig. S1, the skeleton of the trajectory is represented by a series of nodes and edges. In examples S1b-d, the knock-out (KO) has no impact on this skeleton compared to the wild-type (WT). In example S1e, the knock-out modifies the skeleton, in that the locations of the nodes change. However, the adjacency matrix does not change and the two skeletons represent isomorphic graphs, so that the skeleton structure is preserved.

A common skeleton structure can also be used if the null hypothesis of the `topologyTest` is rejected. The availability of a mapping between lineages means that the next steps of the workflow can be conducted as if we had failed to reject the null hypothesis, as done in Fig. S1e. The KRAS (supplementary section S-2.8) case study presents an example of this. Even if the null is rejected by the `topologyTest` and separate trajectories must be fitted for each condition, a common skeleton structure can still be used to map between trajectories. In cases where no common skeleton structure exists, such as Fig. S1f, no automatic mapping exists. Differential abundance can be assessed but requires a manual mapping. Differential expression can still be conducted as well.

### Testing for differential progression

The `progressionTest` requires that a (partial) mapping exist between trajectories, or that a common trajectory be fitted across conditions. If the mapping is only partial, we restrict ourselves to the mappable parts of the trajectories (i.e., subgraphs).

For a given lineage $l$, we want to test the null hypothesis that the pseudotimes along the lineage are identically distributed between conditions, which we call *identical progression*. Specifically, we want to test that the $l^{th}$ components $G_{lc}$ of the distribution functions $G_c$ are identical across conditions

$$H_0 : G_{lc_1} = G_{lc_2}, \forall (c_1, c_2). \tag{7}$$

We can also test for global differences across all lineages, that is,

$$H_0 : G_{c_1} = G_{c_2}, \forall (c_1, c_2). \tag{8}$$

If $C = 2$, all tests introduced at the beginning of the Methods section can be used to test the hypothesis in Equation (7). If $C > 2$, we need to rely on the classifier test.

If $L = 1$, the hypotheses in Equations (7) and (8) are identical. However, for $L > 1$, the functions $G_c$ are not univariate distributions and this requires different testing procedures.

For $L > 1$, we can use observational weights for each cell corresponding to the probability that it belongs to each individual lineage. Two settings are possible.

- Test the null hypothesis in Equation (7) for each lineage using the Kolmogorov-Smirnov test and perform a global test using the classifier test or the kernel two-sample test.
- Test the null hypothesis in Equation (7) for each lineage using the Kolmogorov-Smirnov test and combine the $p$-values $p_l$ for each lineage $l$ using Stouffer's Z-score method[46], where each lineage is associated with observational weights $W_l = \sum_{i=1}^{n} \mathbf{W}_i[l]$, with $\mathbf{W}_i[l]$ the $l^{th}$-coordinate of the vector $\mathbf{W}_i$. The nominal $p$-value associated with the global test is then

$$p_{glob} \equiv \frac{\sum_{l=1}^{L} W_l p_l}{\sqrt{\sum_{i=1}^{L} W_l^2}}.$$

Note that the second setting violates the assumption of Stouffer's Z-score method, since the $p$-values are not i.i.d. However, this violation does not seem to matter in practice and this test outperforms others, so we set it as default.

## Testing for differential fate selection

The `fateSelectionTest` requires that a (partial) mapping exist between trajectories. If the mapping is only partial, we restrict ourselves to the mappable parts of the trajectories.

For a given pair of lineages $\{l, l'\}$, we want to test the null hypothesis that the cells differentiate between $l$ and $l'$ in the same way between all conditions, which we call *identical fate selection*. Specifically, we want to test that the $l^{th}$ and $l'^{th}$ components of the distribution functions $H_c$ for the weights are the same across conditions

$$H_0 : \forall(c_1, c_2), [H_{lc_1}, H_{l'c_1}] = [H_{lc_2}, H_{l'c_2}]. \quad (9)$$

We can also test for a global difference across all pairs of lineages, that is,

$$H_0 : \forall(c_1, c_2), H_{c_1} = H_{c_2}. \quad (10)$$

Since all variables are multivariate, we cannot use the Kolmogorov-Smirnov test. By default, this test relies on the classifier test with random forests as a classifier.

## Testing for differential expression

The gene expression model does not require a mapping or even a partial mapping between trajectories. Indeed, it can work equally well with a common trajectory, different trajectories, or even partially shared trajectories where some lineages can be mapped between the trajectories for various conditions and others cannot. To reflect this, we consider all $L_{tot}$ lineages together. We introduce a new weight vector for each cell:

$$\mathbf{Z}_i = \{Z_{ilc}\}_{l \in \{1, \dots, L_{tot}\}, c \in \{1, \dots, C\}} \text{ s.t.} \begin{cases} Z_{ilc} = 0, & \text{if } c(i) \neq c \text{ or } l \notin \mathcal{T}_{c(i)} \\ \{Z_{ilc(i)}\}_{l \in \{l : l \in \mathcal{T}_{c(i)}\}} \sim \mathcal{M}(\mathbf{W_i}), & \text{otherwise} \end{cases},$$

where $\mathcal{M}(\mathbf{W}_i)$ is a binary (or one-hot) encoding representation of a multinomial distribution with proportions $\mathbf{W}_i$ as in tradeSeq, i.e. $\mathbb{P}(Z_{ilc(i)} = 1) = \mathbf{W}_i[l]$ if $l \in \mathcal{T}_{c(i)}$, and $l \in Tc(i)$ is an abuse of notation to denote that a lineage belongs to a particular trajectory.

Likewise, we modify the pseudotime vectors to have length $L_{tot}$, such that

$$T_{li} = \begin{cases} 0, & \text{if } l \notin \mathcal{T}_{c(i)} \\ \mathbf{T}_i[l], & \text{otherwise} \end{cases}.$$

We adapt the model from Van den Berge et al.[8] to allow for condition-specific expression. For a given gene $j$, the expression measure $Y_{ji}$ for that gene in cell $i$ can be modeled as:

$$\begin{cases} Y_{ji} & \sim & NB(\mu_{ji}, \phi_j) \\ \log(\mu_{ji}) & = & \eta_{ji} \\ \eta_{ji} & = & \sum_{l=1}^{L_{tot}} \sum_{c=1}^{C} s_{jlc}(T_{li}) Z_{ilc} + \mathbf{U}_i \boldsymbol{\alpha}_j + \log(N_i) \end{cases}, \quad (11)$$

where the mean $\mu_{ji}$ of the negative binomial distribution is linked to the additive predictor $\eta_{ji}$ using a logarithmic link function. The $\mathbf{U}$ matrix represents an additional design matrix, corresponding, for example, to a batch effect, and $N_i$ represents sequencing depth, i.e., $N_i = \sum_j Y_{ij}$.

The model relies on lineage-specific, gene-specific, and condition-specific smoothers $s_{jlc}$, which are linear combinations of $K$ cubic basis functions, $s_{jlc}(t) = \sum_{k=1}^{K} b_k(t) \beta_{jlck}$.

With this notation, we can introduce the `conditionTest`, which, for a given gene $j$, tests the null hypothesis that the smoothers are identical across conditions:

$$H_0 : \forall(c_1, c_2), \forall k, \forall l, \beta_{jlc_1 k} = \beta_{jlc_2 k}. \quad (12)$$

We fit the model using the mgcv package[47] and test the null hypothesis using a Wald test for each gene. Note that, although the gene expression model can be fitted without any mapping, the `conditionTest` only exists for lineages with a mapping for at least two conditions.

Furthermore, rather than attaching strong probabilistic interpretations to *p*-values (which, as in most RNA-seq applications, would involve a variety of hard-to-verify assumptions and would not necessarily add much value to the analysis), we view the *p*-values produced by the **condiments** workflow simply as useful numerical summaries for exploring trajectories across conditions and identifying genes for further inspection.

## Simulation study

The simulation study relies on the dyngen framework of Cannoodt et al.[24] and all datasets are simulated as follows. 1/ A common trajectory is generated, with an underlying gene network that drives the differentiation along the trajectory. 2/ A set of $N_{WT}$ cells belonging to the wild-type condition (i.e., with no modification of the gene network) is generated. 3/ One master regulator that drives differentiation into one of the lineages is impacted, by multiplying the wild-type expression rate of that gene by a factor $m$. If $m = 1$, there is no effect; if $m > 1$, the gene is over-expressed; and if $m < 1$, the gene is under-expressed, with $m = 0$ amounting to a total knock-out. 4/ A set of $N_{KO} = N_{WT}$ knock-out cells is generated using the common trajectory with the modified gene network. 5/ A common reduced-dimensional representation is computed using multidimensional scaling.

We generate four types of datasets, over a range of values of $m$: a simple trajectory with $L = 2$ lineages and $C = 2$ conditions (WT and KO) denoted by $\mathcal{T}_1$; a trajectory with two consecutive branchings with $L = 3$ lineages and $C = 2$ conditions (WT and KO) denoted by $\mathcal{T}_2$; a simple trajectory with $L = 2$ lineages and $C = 3$ conditions (WT, KO, and UP) denoted by $\mathcal{T}_3$ (for the latter case, dyngen Steps 3-4/ are repeated twice, with values of $m$ for KO and $1/m$ for UP); and a trajectory with two consecutive branchings with $L = 5$ lineages and $C = 2$ conditions (WT and KO) denoted by $\mathcal{T}_4$.

For $\mathcal{T}_1$ and $\mathcal{T}_2$, we use values of $m \in \{0.5, 0.8, 0.9, 0.95, 1, 1/0.95, 1/0.9, 1/0.8, 1/0.5\}$, such that at the extremes the KO cells fully ignore some lineages. Values of .95 and 1/.95 represent the closest to no condition effect ($m = 1$), where the effect was still picked out by some tests. For $\mathcal{T}_3$, since the simulation is symmetrical in $m$, we pick $m \in \{0.5, 0.8, 0.9, 0.95, 1\}$. For $\mathcal{T}_4$, we use values of $m \in \{0.1, 0.2, 0.3, 0.5, 1, 2, 3, 5, 10\}$.

We have one large dataset per value of $m$ and per trajectory type. We use these large datasets to generate smaller ones of size $n$, by sampling 10% of the cells from each condition 50 times and applying the various tests on the smaller datasets. The reason for first generating a large dataset and then smaller ones by subsampling instead of generating small ones straightaway are computational: the generation of the datasets is time-consuming and the part that scales with $N_{WT}$ can be parallelized. Hence, it is almost as fast to generate a large dataset than a small one with dyngen. We pick $N_{WT} = 20,000$ (for the large dataset) and thus $n = 2000$.

Since we generate many datasets with true effect ($m \neq 1$) but only one null dataset, the size of $N_{WT}$ for $m = 1$ is doubled to 40,000. To be comparable, the fraction of cells sampled is decreased to 5%, so that $n = 2000$ and we perform 100 subsamplings. Table 3 recapitulates all of this.

To run the **condiments** workflow, we first estimate the trajectories using slingshot with the clusters provided by dyngen. Then, we

**Table 3 | Summary of all simulated datasets**

| Dataset | $N_{WT}$ | | $n$ | $L$ | $C$ | Impacted | Gene Network | Reduced Dimension |
|---|---|---|---|---|---|---|---|---|
| | $m \neq 1$ | $m = 1$ | | | | Regulator | Figure | Representation |
| $\mathcal{T}_1$ | 20, 000 | 40, 000 | 2, 000 | 2 | 2 | B3 | Fig. S3a | Fig. 2a |
| $\mathcal{T}_2$ | 20, 000 | 40, 000 | 2, 000 | 3 | 2 | D2 | Fig. S3b | Fig. 2b |
| $\mathcal{T}_3$ | 20, 000 | 40, 000 | 2, 000 | 2 | 3 | B3 | Fig. S3a | Fig. 2c |
| $\mathcal{T}_4$ | 20, 000 | 40, 000 | 2, 000 | 5 | 3 | B17 | Fig. S3c | Fig. 2d |

The table reports the name, number of cells $N_{WT}$ for values of $m \neq 1$ and $m = 1$, number of conditions $C$, number of lineages $L$, impacted master regulator, and figure numbers for the associated gene network and an example of low-dimensional representation.

run the `progressionTest` and the `fateSelectionTest` with default arguments.

We compare **condiments** to two other methods. **milo**[16] and **DAseq**[17] both look at differences in condition proportions within local neighborhoods, using $k$-nearest-neighbor graphs to define this locality. Then, **milo** uses a negative binomial GLM to compare counts for each neighborhood, while **DAseq** uses a logistic classifier test. Therefore, both methods test for differential abundance in multiple regions. To account for multiple testing, we adjust the $p$-values using the Benjamini-Hochberg[25] FDR-controlling procedure.

We select an adjusted $p$-value cutoff of 0.05, which amount to controlling the FDR at nominal level 5%. For a given dataset, we can look at the results of each test on all simulated datasets for all values of $m$. For each test, the number of true positives (TP) is the number of simulated datasets where $m \neq 1$ and the adjusted $p$-value is less than 0.05, the number of true negatives (TN) is the number of simulated datasets where $m = 1$ and the adjusted $p$-value is greater than or equal to 0.05, the number of false positives (FP) is the number of simulated datasets where $m = 1$ and the adjusted $p$-value is less than 0.05, and the number of false negatives (FN) is the number of simulated datasets where $m \neq 1$ and the adjusted $p$-value is greater than or equal to 0.05. We then examine five measures built on these four variables:

$$\text{True Negative Rate (TNR)} = \frac{TN}{TN + FP}$$
$$\text{True Positive Rate (TPR)} = \frac{TP}{TP + FN}$$
$$\text{Positive Predictive Value (PPV)} = \frac{TP}{TP + FP}$$
$$\text{Negative Predictive Value (NPV)} = \frac{TN}{TN + FN}$$
$$\text{F1-score} = 2\frac{PPV \times TPR}{PPV + TPR}.$$

We also sought to generate incorrect inferences as follows. We generate a dataset of type $\mathcal{T}_1$ with a multiplier effect $m = 0.85$. We use the true pseudotimes and lineage assignment values, as defined by dyngen, and we add increasing noise along two dimensions: pseudotime and lineage assignment. For the latter, we randomly select a proportion $p$ of cells and change their lineage assignments; $p = 0$ means that we use the true lineage assignments, while $p = .5$ means that the lineage assignment of the cells is fully random. For the former, we add random Gaussian noise such that, for a cell $i$ with pseudotime $\mathbf{T_i}$, we return a new pseudotime $\tilde{\mathbf{T}}_\mathbf{i} = \mathbf{T_i} \times \mathcal{N}_2(\mathbb{I}_2, sd \times \mathbb{I}_2)$. We choose $p \in \{0, .1, .2, .3, .4, .5\}$ and $sd \in [0:40]/10$. For each value of $p$ and $sd$, we did 100 runs and selected the median $p$-value.

**Version of tools used**
The following version have been used:
- condiments: 1.4.0
- tradeSeq: 1.10.0
- slingshot: 2.4.0
- monocle3: 1.2.9
- DASeq: 1.0.0
- miloR: 1.4.0
- dyngen: 1.0.3

**Reporting summary**
Further information on research design is available in the Nature Portfolio Reporting Summary linked to this article.

## Data availability
The results from this paper can be fully reproduced by following along the vignettes at https://hectorrdb.github.io/condimentsPaper. Code to reproduce the datasets used for the simulation study, as well as processed versions of all four datasets used in the case studies, augmented by metadata, are in particular available at https://github.com/HectorRDB/condimentsPaper/tree/main/data. Functions to recreate the processed versions, using raw counts obtained from GEO (**TGFB** dataset: GSE114687 - https://www.ncbi.nlm.nih.gov/geo/query/acc.cgi?acc=GSE114687, **TCDD** dataset: GSE148339 - https://www.ncbi.nlm.nih.gov/geo/query/acc.cgi?acc=GSE148339, **KRAS** dataset: GSE137912 - https://www.ncbi.nlm.nih.gov/geo/query/acc.cgi?acc=GSE137912, **Fibrosis** dataset: GSE135893 - https://www.ncbi.nlm.nih.gov/geo/query/acc.cgi?acc=GSE135893) are also provided.

## Code availability
The **condiments** workflow is available as an R package from GitHub (https://github.com/HectorRDB/condiments[48]) and through the Bioconductor Project (https://www.bioconductor.org/packages/release/bioc/html/condiments.html). All the methods to test for equality of two (or $k$) distributions were implemented for use by others in an R package called Ecume, available through CRAN (http://cran.r-project.org) and that can be explored at https://hectorrdb.github.io/Ecume.

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

## Acknowledgements

This work was supported by the NIH R01 R01DC007235. KVdB was a postdoctoral fellow of the Belgian American Educational Foundation (BAEF) and was supported by the Research Foundation Flanders (FWO), grants 1246220N, G062219N, and V411821N.

## Author contributions

H.R.B. and K.V.D.B. developed the methods under S.D.'s supervision. H.R.B. ran the experiments and designed the simulation studies, with

directions from K.S. and S.D. H.R.B. wrote the first draft of the manuscript, and H.R.B., K.V.D.B., K.S. and S.D. provided comments and helped in the revisions.

## Competing interests

The authors declare no competing interests.
