## [Peer Review File · Nature Communications]

Reviewers' Comments:

Reviewer #1:

Remarks to the Author:

I've reviewed the response, and it seems the authors have addressed my previous questions to the best of their ability. While some limitations remain inherent in their framework, it's worth noting that the authors have been transparent about these issues in their manuscript. Given this acknowledgment, I would recommend accepting the paper.

Regarding my first previous question (Q1), the authors have clarified that batch correction has already been performed.

As for my second previous question (Q2), the authors provided additional insights into their analysis.

Concerning my third previous question (Q3), the authors acknowledge that their test doesn't yield correct p-values. Although they control the false discovery rate (FDR) after increasing logFC in their simulation setting, it's important to note that this FDR control seems to stem from conservatively adjusting p-values. Considering that their previous paper, tradeSeq, was accepted for publication, I believe this paper also holds merit. It's crucial to recognize the inherent difficulty in achieving precise statistical interpretation in this particular topic.

In summary, while some concerns persist, the authors' responsiveness and the acknowledged limitations, as well as the broader challenges in the field, make accepting this paper a reasonable choice.

Reviewer #2:

Remarks to the Author:

Authors have addressed my previous comments. I have one more comment on the following statement.

"Other methods have been presented that can generate p-values with probabilistic interpretation [33, 34]. However, current methods are restricted to only one lineage, or scale poorly beyond a thousand cells, showing the need for further work in this domain."

The reference biorxiv Paper 33 has been recently published in Nature Communications (Hou et al. PMID: PMC10638410). It seems the published version has the similar goal as the current paper and can analyze multiple samples with many cells. It is unclear to me about the significant improvement or novelty of condiments comparing to lamian in that paper. Authors should do some comparisons on the presented tasks.

Response to reviewers: Trajectory inference across multiple conditions with **condiments**

December 18, 2023

1 Reviewer 1

I've reviewed the response, and it seems the authors have addressed my previous questions to the best of their ability. While some limitations remain inherent in their framework, it's worth noting that the authors have been transparent about these issues in their manuscript. Given this acknowledgment, I would recommend accepting the paper.

Regarding my first previous question (Q1), the authors have clarified that batch correction has already been performed.

As for my second previous question (Q2), the authors provided additional insights into their analysis.

We thank the reviewer for their constructive comments and their help in making this manuscript publication-worthy.

*Concerning my third previous question (Q3), the authors acknowledge that their test doesn't yield correct p -values. Although they control the false discovery rate (FDR) after increasing logFC in their simulation setting, it's important to note that this FDR control seems to stem from conservatively adjusting p -values. Considering that their previous paper, *tradeSeq*, was accepted for publication, I believe this paper also holds merit. It's crucial to recognize the inherent difficulty in achieving precise statistical interpretation in this particular topic.*

Producing p -values with precise probabilistic interpretation is indeed challenging in such settings. Testing against a non-zero logFC is a routine approach in differential gene expression analysis to increase the biological significance of the results, rather than specifically induce conservativeness. However, as stated in the discussion section, we do not claim that our approach leads to p -values with a clear probabilistic interpretation. New methods accounting for pre-processing steps preceding the condiments workflow, e.g., normalization and dimensionality reduction, could help in this regard.

2 Reviewer 2

Authors have addressed my previous comments. I have one more comment on the following statement.

"Other methods have been presented that can generate p -values with probabilistic interpretation [33, 34]. However, current methods are restricted to only one lineage, or scale poorly beyond a thousand cells, showing the need for further work in this domain."

*The reference *bioRxiv* Paper 33 has been recently published in *Nature Communications* (Hou et al. PMID: PMC10638410). It seems the published version has the similar goal as the current paper and can analyze multiple samples with many cells. It is unclear to me about the significant improvement or novelty of condiments comparing to *lamian* in that paper. Authors should do some comparisons on the presented tasks.*

We thank the reviewer for their help in improving our manuscript. Regarding their final comment, we have updated the reference to the published version of the Hou et al. (2023) paper. This version, just published in November 2023 while our paper was under revision, is significantly different from the previous *bioRxiv* version and itself does not include the most recent version of condiments that proposes new tests, including distinct [1], that can handle multiple samples per condition.

References

- [1] Simone Tiberi, Helena L. Crowell, Pantelis Samartsidis, Lukas M. Weber, and Mark D. Robinson. distinct: A novel approach to differential distribution analyses. *The Annals of Applied Statistics*, 17(2):1681 – 1700, 2023. doi: 10.1214/22-AOAS1689. URL <https://doi.org/10.1214/22-AOAS1689>.